# Modified Polymer Surfaces: Thin Films of Silicate Composites via Polycaprolactone Melt Fusion

**DOI:** 10.3390/ijms23169166

**Published:** 2022-08-15

**Authors:** Eva Skoura, Peter Boháč, Martin Barlog, Helena Palková, Martin Danko, Juraj Šurka, Andreas Mautner, Juraj Bujdák

**Affiliations:** 1Institute of Inorganic Chemistry, Slovak Academy of Sciences, Dúbravská cesta 9, SK-845 36 Bratislava, Slovakia; 2Centre of Excellence for Advanced Materials Application, Slovak Academy of Sciences, SK-845 11 Bratislava, Slovakia; 3Polymer Institute, Slovak Academy of Sciences, SK-845 41 Bratislava, Slovakia; 4Earth Science Institute, Slovak Academy of Sciences, Ďumbierska 1, SK-974 11 Banská Bystrica, Slovakia; 5Polymer and Composite Engineering (PaCE) Group, Department of Materials Chemistry, Faculty of Chemistry, University of Vienna, Währinger Str. 42, 1090 Wien, Austria; 6Department of Physical and Theoretical Chemistry, Faculty of Natural Sciences, Comenius University in Bratislava, SK-842 15 Bratislava, Slovakia

**Keywords:** methylene blue, organoclay, polymer melt, surface modification, surface properties

## Abstract

Polymer/layered silicate composites have gained huge attention in terms of research and industrial applications. Traditional nanocomposites contain particles regularly dispersed in a polymer matrix. In this work, a strategy for the formation of a composite thin film on the surface of a polycaprolactone (PCL) matrix was developed. In addition to the polymer, the composite layer was composed of the particles of saponite (Sap) modified with alkylammonium cations and functionalized with methylene blue. The connection between the phases of modified Sap and polymer was achieved by fusing the chains of molten polymer into the Sap film. The thickness of the film of several μm was confirmed using electron microscopy and X-ray tomography. Surfaces of precursors and composite materials were analyzed in terms of structure, composition, and surface properties. The penetration of polymer chains into the silicate, thus joining the phases, was confirmed by chemometric analysis of spectral data and changes in some properties upon PCL melting. Ultimately, this study was devoted to the spectral properties and photoactivity of methylene blue present in the ternary composite films. The results provide directions for future research aimed at the development of composite materials with photosensitizing, photodisinfection, and antimicrobial surfaces.

## 1. Introduction

Polymer nanocomposites represent a very wide range of materials characterized by dispersed nanoparticles in a polymer matrix including clay-based or layered silicate/polymer nanocomposites [1,2]. The topic of clay/polymer nanocomposites has been analyzed in several reviews [2,3,4]. There are various strategies for preparing polymer nanocomposites, such as in-situ polymerization from a liquid phase with dispersed nanoparticles, sol-gel methods, melt extrusion, intercalation from solution and solution casting, syntheses using supercritical fluids, and others [4]. Numerous applications of composites relate to improved mechanical and barrier properties and thermal stability [1,2]. Melt intercalation is a standard method to synthesize polymer nanocomposites, which is compatible with industrial processes, and environmentally friendly without the use of organic solvents. This method was also applied in this work. In the case of layered nanoparticles, three types of composites or phases can be obtained in terms of structure: non-intercalated, intercalated and exfoliated (or delaminated) [1,2].

In general, the surface properties of composites are only slightly affected by the introduction of nanoparticles given that their concentration in the polymer matrix does not reach high values. Most of the particles dispersed in the polymer bulk are inactive with respect to the interactions at the surface of polymer composite. Yet, there is an increasing demand for materials with special properties and extra functionalities relating to surface properties. However, a large concentration of a modifier is needed to change the surface properties which results in loss of some beneficial mechanical properties typical for pure polymers. To avoid this, modification selectively at or close to the surface is required and actually might be limited to a thin layer in the range of nano- to micrometers. For instance, polyurethane membranes were modified with an organoclay (OC) focusing on surface properties [5]. The membranes exhibited antimicrobial properties with a remarkable ability to reduce the growth of bacterial biofilms. A similar strategy based on polymer fusion to achieve surface modification is applied in this work. Polymer melting takes place at the interface and in contact between two phases, a thin OC film, and the molted polymer phase. The OC was derived from synthetic saponite (Sap), modified with a cationic surfactant, and functionalized with methylene blue (MB). Polycaprolactone (PCL), a polymer with a low melting point [6,7] was used as a model system. Clay/PCL nanocomposites have been studied extensively, and most of the main features of these systems are well summarized in a review article [7]. The properties of PCL composites depend much on the method of preparation (melt extrusion, monomer polymerization, solution casting) [6,7,8] and hence the possible applications of PCL nanocomposites are very broad. They are often used as additives to other polymers to improve their mechanical or thermal properties [6,9] and exhibit antimicrobial properties, either alone [10,11] or in combination with other active nanoparticles [12,13] or molecules [14]. The biocompatibility of clay/PCL nanocomposites has been reported [15] and phenomena of slow release of drug molecules or bioactive substances from PCL composites have also been investigated [14,16,17].

In addition to the usual traditional polymer composites, it should be feasible to prepare hybrid systems with a locally increased concentration of dispersed particles located on the surface or near the surface of the polymer. The objective of this work was to prepare PCL membranes with a surface modified with an OC layer. PCL melting in contact with a thin OC film was the method used for the polymer surface modification. The partial penetration of the polymer chains from the melt into the OC phase allowed for strong bonding between the two phases, but joining the two phases together should not change the important properties of the polymer bulk phase. Compositional and structural characterizations of the surface phase were performed. One of the motivations for preparing surface-modified composites was to develop functional materials with surfaces exhibiting antimicrobial and antibiofilm activity, which can be achieved by the presence of a photosensitizer as an active component. Therefore, PCL nanocomposites containing the dye methylene blue were also developed.

## 2. Results and Discussion

The PCL sample used in this work is a commercial product with a relatively high molar mass (~10^5^ mol mol^−1^) but exhibits all the typical properties of PCL, such as especially the low melting temperature. Detailed characterization of the polymer and some discussion of its parameters are given in the Appendix A.

The synthesis of composites is described in more detail in the Section 3. For the sake of brevity, only the schematic description is mentioned here. In the first step, Sap colloidal systems were prepared, which were modified with an organic surfactant (hexadecyltrimethylammonium, HDTMA) to create hydrophobic particles. Colloids with MB-functionalized particles were also prepared. Thin films with a thickness of several micrometers were prepared by vacuum filtration. Polymer membranes with a composite layer were prepared by melting PCL on the surface of these films. It was the low melting point that enabled easy preparation of a thin film of composite on the surface of polymer membranes. Abbreviations used to identify samples of organoclays (OC, MB1, MB2,..., MB10) and their composites with the polymer (PCL/OC, PCL/MB1, PCL/MB2,…, PCL/MB10) are explained in Section 3.2 and Section 3.3. The properties of the composite films were the main object of this study and included film thickness (Section 2.1), surface hydrophobicity (Section 2.2), crystalline phase structure (Section 2.3), chemical composition, and the presence of functional groups (Section 2.4 and Section 2.5). In the case of MB-containing films, optical properties such as light absorption and luminescent properties were also characterized (Section 2.6).

### 2.1. Thickness of the Films

Scanning electron microscopy (SEM) images were recorded to characterize the topography of the surface and the thickness of OC film on the PCL membrane (Appendix A). Neat PCL pellets exhibited a smooth surface even at 5000× magnification (Appendix A). As opposed to pure PCL, the PCL/MB10 membrane exhibited a more irregular surface caused by the OC particles present on the top of the surface (Appendix A). Investigation of the cross-section of the composite membranes allowed for the analysis of the material in a profile perpendicular to the surface (Figure 1a,b). SEM was able to capture the PCL/MB10 composite phase on the membrane surface in the form of a thin film. This film could be distinguished from the pure phase of PCL. The brittleness of the composite on the surface led at some sites to disruption of the contact between the polymer and the composite phase. The actual film thickness was about 5 μm and maintained a decent uniformity that varied in the range of ±0.5 μm. A similar value was confirmed using X-ray tomography (Figure 1c,d), although the thickness value was close to the detection limit of the method. Statistical analysis yielded a distribution range between 2–12 μm (Figure 1d) while the mean value of (5.67 ± 0.05) μm was in good agreement with the values observed using SEM (Figure 1b).

### 2.2. Hydrophobicity of the Surface of Prepared Materials

The surface properties of selected organoclays (OC, MB4, MB10) and their composites were investigated by employing water contact angle measurements (Table 1). Organoclays OC and MB4 exhibited less hydrophilic surfaces with contact angles slightly above 60°. The presence of HDTMA cations in OC reduced the hydrophilicity of Sap. A similar effect of the surfactant was observed for analogous organoclays [5]. The functionalization of OC with a small amount of MB led to an insignificant change in the contact angle (at a 0.05 probability level) (Table 1). Functionalization with a large amount of MB resulted in a less hydrophobic surface exhibiting a water contact angle of 47 ± 5°. One possible explanation is a higher polarity of the MB cations compared to the HDTMA surfactant. The water contact angle on the MB10 sample was significantly lower than those of all other specimens, proven statistically at a significance level of 0.001. Neat PCL and all its composites exhibited contact angles above 60° similar to the hydrophobic organoclays (Table 1). The presence of OC did not have a considerable effect on the contact angle values. The composite sample with the highest MB amount (PCL/MB10) did not exhibit reduced hydrophobicity. This indicates, that MB10 and PCL/MB10 were different in the compositions of their surfaces. One of the explanations is based on the assumption that the melted polymer penetrated the phase of the MB10 organoclay film and turned the surface more hydrophobic. Thus the contact angle measurements provide indirect but valuable evidence of the penetration of polymer chains into the film of an organoclay phase.

### 2.3. Crystallinity by X-ray Diffraction

The expansion of the interlayer spaces of Sap by intercalation with surfactant, dye, or polymer can be detected by X-ray diffraction (XRD). Polymer composite membranes were compared with films of organoclays and unmodified Sap (Figure 2). The expansion of the basal spacing *d*_001_ increased from 1.38 nm in Sap to about 1.49 nm in HDTMA/Sap. This change can be assigned to the intercalation of large HDTMA cations between Sap layers, exchanging hydrated exchangeable cations and free water molecules. A further increase of the spacing (1.59 nm) was observed for the sample MB10, containing the largest amount of MB in addition to HDTMA. This trend confirmed the successful intercalation of both HDTMA and MB. The *d*_001_-values indicated the presence of a mixed monolayer/bilayer phase (1.36 nm/1.76 nm) of alkyl chains of HDTMA cations present in organoclay films [18].

PCL is a semicrystalline material and its crystalline phase can be detected in the range of 15–30° (2*θ*) [19]. Three reflections at 21.4°, 22°, and 23.7° (Figure 2b) were attributed to (110), (111), and (200) planes of the polymer, respectively [20]. No reflection from the pure polymer occurred at lower angles. PCL crystalline phase was also detected in the composite samples. The peaks assigned to the PCL phase did not significantly change in the XRD patterns of composite materials. The detection of the polymer phase in the composite materials can in fact be caused by the actual presence of PCL in the organoclay films, but a large penetration depth of X-rays beneath the films reaching the pure polymer phase should also be considered. Basal reflections assigned to organoclay in the XRD patterns of composites PCL/OC or PCL/MB10 were not significantly changed compared to the pure organoclay films OC and MB10 (Figure 2a). The polymer chains most probably did not intercalate in the interlayer spaces of the organoclay phase. The intercalation would lead to a measurable expansion of the interlayer spaces and shifts of the basal reflections to lower angles. However, an opposite trend was observed with a shift of *d*_001_ to slightly lower values (1.41 and 1.45 nm) which can be explained in terms of the release of adsorbed volatile compounds from organoclay samples upon the thermal treatment when interacting with the melted polymer. In addition, the molar mass of the polymer is relatively high (Appendix A), which affects the mobility of the melted polymer chains, and therefore may slow or limit the penetration of the two phases thus preventing intercalation [21]. Melting of the polymer at the interface with the organoclay film was not promoted by any active phase mixing but depended only on the thermal motion of the polymer chains at the interface. Exfoliation or intercalation of the polymer chains in the organoclay phase was not expected. On the other hand, the linkage and fusion of the phases have been clearly demonstrated experimentally. The absence of intercalation has also been reported for most PCL/organoclay composites prepared by melt extrusion [6,22,23,24] and the intercalated phase was detected only rarely [25,26]. In a few papers, expansion of the interlayer spaces was claimed but only poor X-ray reflections could be recognized [6]. Some studies have relied on transmission electron microscopy analysis, but interpretations were rather subjective and depended on the quality of the TEM specimens [6,23]. More often, exfoliated or intercalated phases were observed for in situ-intercalative monomer polymerization [27]. Overall, XRD measurements did not provide any evidence of PCL intercalation between Sap layers but indicated the possible presence of polymer chains in the phase of organoclay film. Nevertheless, other methods had to be used to support this assumption.

### 2.4. Chemical Functionality by Infrared Spectroscopy

Attenuated total reflectance infrared (ATR IR) spectroscopy was carried out to detect the presence of individual components and functional groups on the surface of the prepared samples. The bands assigned to vibrations of characteristic groups of both Sap and HDTMA components can be found in the organoclay spectrum (Figure 3a). Their identification was confirmed by comparison with the spectra of the single components or precursors. Stretching vibration (ν) of the structural OH group occurred at 3670 cm^−1^. Structurally this means that the OH group is shared by three octahedra with Mg atoms in central positions (MgMgMgOH). The band has a low intensity in the ATR spectrum of Sap; however, after modification with HDTMA and MB, it changed to a more intensive and sharp band. Such a trend could be attributed to the removal of a significant part of the water molecules that had formed the hydration shells of exchangeable cations and filled interlayer spaces. The presence of H_2_O is reflected in the spectra by a wide complex band near 3440 cm^−1^. After modification with organic cations, most of water molecules were removed from the samples. This process decreased the overall intensity of νOH from H_2_O. Subsequently, the band at 3670 cm^−1^ became more pronounced for OC and MB10. Two intense bands at 2923 and 2847 cm^−1^ were attributed to the C-H stretching vibrations of the aliphatic chains of HDTMA cations. Corresponding C-H bending vibrations were present between 1500–1300 cm^−1^. An intense band at a wavenumber of 965 cm^−1^ indicates the Si-O stretching vibration of the Sap component. The bands at 656 and 421 cm^−1^ correspond to the bending vibrations of structural OH (MgMgMg-OH) and Si-O-Mg bonds present in Sap layers [28]. Despite the very low content of MB in the samples it could be detected based on the band at 1600 cm^−1^ in the MB10 spectrum. The band was assigned to the C=C and C=N stretching of the heterocycle of MB revealing the presence of this dye in the sample [29,30] and it is one of the most intensive bands in the spectra of the MB molecule (Figure 3b). IR spectra of composite materials and PCL were compared. In the spectrum of neat PCL, two bands at 2928 and 2851 cm^−1^ were attributed to the stretching C-H vibrations of methylene-oxygen (CH_2_-O) and symmetric methylene groups (CH_2_-), respectively. The bands in this region may also be overlapping with those of HDTMA present in organoclays. Hence, the lower wavenumber region is better suited to detect the presence of PCL. A characteristic feature of this polymer was a strong band at 1725 cm^−1^ corresponding to the C=O bond vibrations [31]. The presence of this band was detected in the spectra of modified PCL membranes (PCL/OC and PCL/MB10) and it is important evidence of the polymer diffusion to the surface of the composites. The penetration of IR light in the ATR techniques is limited to about 2 μm [32], which is less than the thickness of the composite film (~5 μm, see Figure 1 and discussion in Section 2.1). However, the ν(C=O) band does not dominate the spectra of the composites, so it could be assumed that the diffusion of polymer occurred only partially and probably in low content into OC or MB10 films. It is also worth noting that the most intensive band of MB (at 1600 cm^−1^) was still visible in the spectra of composite PCL/MB10.

The chemometric method multivariate curve resolution—alternating least squares (MCR) was used to decompose the matrix of measured spectra for semi-quantitative estimation of the occurrence of signals of individual components in more complex materials. Decomposition of the spectral matrix yielded the spectra of the three major components, which corresponded to pure PCL, MB, and OC, and are hereinafter referred to as PCL_MCR_, MB_MCR_, and OC_MCR_. The very good agreement of the calculated spectra of the components (Appendix A) with the measured spectra of the pure substances (Figure 3) confirms the relevance of the results of this method. The values corresponding to the occurrence of the signals of the individual components PCL_MCR_, MB_MCR_, and OC_MCR_ in the measured spectra, i.e., their arbitrary concentrations calculated by the MCR method, are given in Table 2. The values were normalized to 100% with respect to the strongest signal of a given component in the group of the measured spectra. For example, MB_MCR_ logically 100% coincides with the measured spectrum of pure dye. Since the dye contents in the MB10 organoclay and its composite PCL/MB10 sample are relatively small, only about 5% and 2% of the MB_MCR_ signal were present in these samples. Low MB_MCR_ signals (<2%) in the spectra of some samples can be explained by the accuracy limits of the method. The strongest OC_MCR_ signal was observed for OC and PCL/OC samples, which confirms that the surface of the PCL/OC composite is formed mainly by the OC component. A significantly smaller proportion of this component was found for the MB10 sample (94%), which was compensated by the occurrence of MB_MCR_ (5%). The most important results were obtained for the PCL_MCR_ signal. As expected, the strongest signal was recorded for neat PCL. On the other hand, almost 20% of PCL_MCR_ occurred in the measured spectra of both PCL/OC and PCL/M10 which supports the hypothesis of PCL melt fusion into the phase of organoclay film considering a limited penetration depth of IR light into the sample [32].

### 2.5. Surface Elemental Composition by X-ray Photoelectron Spectroscopy

X-ray photoelectron spectroscopy (XPS) was used to analyze the elemental surface composition of composite materials. The measurements provided full-range spectra, which are shown in Appendix A. In alignment with increasing amounts of MB/OC, the contents of both nitrogen and sulfur increased accordingly. Furthermore, the ratio between C and O reversed from 3.5 for PCL, via 1.1 for PCL/MB4, to 0.7 for PCL/MB10. This confirmed the successful introduction of MB at the surface of the films.

Similar to the other analytical methods applied, the attention was focused on the presence of PCL on the surface of the composite membranes. The main tool for polymer identification was the selective detection of the polymer functional groups occurring only in PCL and being absent in other components. The C1s signal was chosen although carbon is present in all organic components (PCL, MB, HDTMA), as it provides the most information about the chemical environment. The common type of C-C bonds along with C-H and C=C ones that occur in each of these components give a signal at ~285 eV. On the other hand, the bands of the C1s signal assigned to C-O and C=O bonds are found at higher energies (~286 eV, ~288.5 eV) and are expected only for PCL, as neither HDTMA nor MB contain oxygen in their molecules. The chemometric analysis using the MCR method clearly confirmed two spectral components (Figure 4a). Only component-1 includes a prominent band found at 288.9 eV, which can be identified with the C=O bonds present in the PCL chains. The band at 285.5 of component-2 may represent the C-N or C-S bonds occurring in HDTMA and/or MB. Indeed, this component has a dominant occurrence, especially in the samples that do not contain PCL (OC, MB10) (Figure 4b). In addition, the relative occurrence of the component-2 signal increased with the amount of MB on the surface of the composite material. While the dominant signal in the spectrum of the PCL/MB4 sample represents also C=O bonds specific to PCL, as the amount of MB increased, the proportion of this component decreased and component-2 predominated in the PCL/MB10 sample. The amount of MB was relatively small and, therefore, this change may not only indicate a larger signal from MB, but also a smaller amount of PCL. The high amount of MB may reduce the hydrophobicity of MB10 as confirmed by the contact angle measurements (Table 1 and discussion in Section 2.2), which brought about a reduction in PCL penetration into the organoclay film. The most interesting result is the fact that component-1, including the PCL selective bands, does not occur or only negligibly occurs in the spectra of OC and MB10 samples (Figure 4b). This is another piece of evidence supporting the assumption that the band at 288.9 eV can only be identified with PCL. Although X-rays penetrate surfaces to a depth of a few micrometers, only the electrons on the surface are active for XPS. The signal is generally limited to a few nanometers depth, which is a perfectly selective probe for our purpose since the thickness of the composite layer was on average about 5 μm (see Figure 1 and discussion in Section 2.1). Thus, the XPS results confirm the presence of the polymer in the composite layer and outer surface.

### 2.6. Optical Properties of Colloidal Precursors, Organoclay Films, and Polymer Composites

One main objective of this study was to develop functional composite materials with surfaces containing methylene blue, which was previously demonstrated to be a photosensitizer [33]. Such materials were prepared from the colloidal precursors based on Sap nanoparticles, modified with HDTMA, and functionalized with MB. The syntheses of organoclay films and PCL composites were performed in the same way as those without MB. The basic optical properties were characterized for the materials functionalized with MB mainly based on spectroscopy in the visible region.

#### 2.6.1. Colloidal Precursors

Information on the properties of MB in hybrid colloidal suspensions of functionalized organoclays was obtained from absorption spectra in the visible region (Appendix A). MB monomers characterized by an absorption band at 670 nm were observed in all samples. However, the colloids with higher concentrations of MB contained also significant amounts of H-dimers (610 nm) and H-aggregates (<600 nm) [34]. MB bulk concentrations were relatively low (Table 3). However, MB cations were concentrated by adsorption on the surface of the Sap particles and surface concentrations exceeded bulk concentrations by several orders of magnitude. Therefore, molecular aggregates occurred even in the samples with the lowest MB concentrations. As the dye concentration increased from MB5 to MB9, the proportion of the band corresponding to the H-dimers (610 nm) increased (Appendix A). In the spectrum of the MB9 sample, the bands of monomers and H-dimers had approximately the same intensities and the relative highest fraction of H-dimers was observed for MB10.

#### 2.6.2. Films of Functionalized Organoclays and Nanocomposite Membranes

The colloidal precursors of organoclays were used to prepare thin films trapped on the surface of filtration membranes. The intensity of the blue color (Figure 5, top) reflects the concentration of the dye increasing in the sample series MB1-MB10 (Appendix A). Ultraviolet-visible (UV-Vis) absorption spectra of organoclay films (Figure 6a) were calculated from diffuse reflectance spectra which therefore provide only qualitative information about the optical properties of the samples. The spectral bands were relatively broad, which was caused by light scattering from the surface of solid films. Nevertheless, it was possible to identify different forms of MB. Due to the low concentrations of the dye in the MB1-MB4 samples, only dye monomers (≈650 nm) were detected. MB monomers were also clearly identified in the samples with a medium dye concentration (MB5-MB7). In the samples with the highest MB concentration (MB8-MB10), the band of monomers broadened to higher wavelengths (680 nm), indicating the presence of J-dimers. The formation of higher J-aggregates was indicated by absorption near 760 nm [35]. The presence of H-dimers (590 nm) started in the sample with a moderate dye concentration (MB6). The proportion of larger H-aggregates that absorbed light at the lowest wavelengths also increased with the dye concentration, reaching the maximum for the MB10 sample with a band at ≈547 nm.

Fluorescence of the films (Figure 6b,c) reflected the dye concentration in the samples, which manifested itself in two opposite trends. Initially, the amount of fluorophore increased with dye concentration for the series of MB1-MB4 samples. The emitted light exhibited lower wavelengths (650 nm) than expected for this dye from the literature (680 nm) [36]. A similar phenomenon was observed for other cationic dyes and explained by the deformation of chromophoric skeletons of the intercalated dye molecules [37]. A significant shift in the emission maximum to higher wavelengths and only a small change in emission intensity were observed with a further increase in dye concentration from the sample MB4 to MB5 (Figure 6c). A radical decrease in fluorescence followed by a regular shift of the emission maximums to higher wavelengths occurred with a further increase in dye concentration (MB6-MB10). These phenomena can be attributed to the concentration quenching caused by the migration and transfer of excitation energy, part of which ended up in inactive molecular aggregates [38]. Overall, the samples with low to medium dye concentrations (MB4 and MB5) exhibited optimal fluorescence properties. The color of the PCL/MB membranes reflected the concentration of the dye (Figure 5). Similar changes in spectra were observed for the PCL composite membranes containing MB (Figure 7 and Figure 8). Light scattering from the optically heterogeneous surface was reflected in the occurrence of broad bands in absorption spectra of the composites (Figure 7), even in the case of the sample with a low MB content (PCL/MB4). On a broad background, the shoulders were assigned to monomers (660 nm), H-dimer (610 nm), larger H-aggregates (560 nm), and J-aggregates (760 nm) [35]. H-aggregates prevailed in the sample with the highest MB content (PCL/MB10).

The fluorescence spectra of PCL membranes showed a trend reflecting the MB concentration (Figure 8) similar to organoclay film precursors (Figure 6b,c). At lower concentrations, the fluorescence intensity increased with the dye concentration, achieving the maximum for MB4 (Figure 8). At higher concentrations, the presence of aggregates led to the quenching of fluorescence and the shift of maximal emission to partially higher wavelengths. The PCL/MB4 composite exhibited the highest emission, but MB5 with a slightly higher amount of the dye was the most emissive sample in the series of precursor films. Fluorescence quantum yields (FQY) confirmed similar properties of both series. PCL/MB4 and MB4 exhibited the highest values (2.5% and 2.1%, respectively). The increase of MB concentration led to a reduction of the FQY values (0.6%, 0.8%, 1.2% and 0.4% for PCL/MB5, PCL/MB6, MB5 and MB6, respectively). These values are at the same magnitude observed for dye solutions (~2%) [39].

## 3. Materials and Methods

### 3.1. Materials

Synthetic Na^+^-saponite (Sap), product name Sumecton SA, purchased from Kunimine Industries Co., Tokyo, Japan, was used without further purification, and information on this silicate has been published elsewhere [40,41,42]. According to the manufacturer, the average particle diameter of Sap is about 20–40 nm. HDTMA bromide (CAS No: 57-09-0) with *M* = 364.45 g mol^−1^ was purchased from Merck KGaA (Darmstadt, Germany). MB hydrate (CAS No: 122965-43-9) was purchased from Sigma Aldrich (St. Louis, MO, USA) and used as received. Commercial PCL polymer for technical use was purchased in the form of solid beads from Didaktik Electronic s.r.o., Slovakia (manufacturer Perstop, Perstorp, Sweden). Isopropanol (iPOH) for spectroscopy use was purchased from Centralchem (CAS No 67-63-0). All solutions and colloids were prepared using Millipore deionized water or organic solvent.

### 3.2. Preparation of Organoclay Films on Teflon Filtration Membranes

A stock colloidal dispersion of Sap (1.0 g L^−1^) was prepared by dispersing Sap powder in water under stirring for 24 h. The HDTMA stock solution (6.4 × 10^−4^ mol L^−1^) was prepared in a mixture of iPOH/water (1:1, *v/v*) and diluted to get a final concentration of 3.2 × 10^−4^ mol L^−1^. Organoclays (OC) were prepared by mixing the HDTMA solution and Sap dispersion using the *n_HDTMA_/m_Sap_* ratio of 0.8 mmol g^−1^. MB was dissolved in water and the concentration was determined by UV-Vis spectroscopy considering the molar absorption coefficient of the dye *ε*_664_ = 9 × 10^4^ mol^−1^ L cm^−1^ [43]. The MB concentration was adjusted to 8 × 10^−5^ mol L^−1^ and diluted solutions were prepared. The solutions were added to the colloids to get systems with variable concentrations of the dye (Table 3). 20 mL colloidal dispersions were filtered via Omnipore PTFE membranes (pore size 0.1 μm, Omnipore^TM^, diameter 25 mm). A photograph of the films containing MB is shown in Figure 5, top. The deposited films had a diameter of about 16 mm and a Sap content of ~4 mg cm^−2^. For organoclays without MB, either in the form of colloids or films, the abbreviation OC is used. For organoclay samples with MB, the abbreviations MB1-MB10 are used reflecting the MB amount present (Table 3).

### 3.3. Preparation of PCL Membranes

Thin disc-shaped membranes of PCL were prepared from PCL beads (≈1.23 g) by hot-pressing at 80 °C. The PCL discs had an approximate diameter and thickness of 1.4 ± 0.2 cm and 0.7 ± 0.2 mm, respectively. In the next step, organoclay films deposited on PTFE membranes were covered with the PCL discs and melted in an oven at 80 °C for 30 min. The melting led to the fusion of PCL into the organoclay film yielding the connection of both phases. After cooling, polymer discs with joint organoclay films were peeled off from the PTFE membrane. For the composite without MB, the abbreviation PCL/OC will be used. For composites with MB (Figure 5, bottom), the abbreviations PCL/MB1, PCL/MB2,..., and PCL/MB10 will be used.

### 3.4. Methods

Details on the methods for basic characterization of PCL such as molar mass determination and thermal analysis are provided in the Appendix A. X-ray diffraction (XRD) patterns of OC films and PCL composite membranes were measured in reflection mode using an X-ray diffractometer X’Pert PRO (PANalytical B.V. Almelo, The Netherlands), equipped with CuK_α_ (*λ*_α1_ = 1.54060 Å) radiation and operating at 45 kV and 40 mA. The patterns were scanned in the 2*θ* range of 2–15° or 2–30° with a scanning step of 0.026° (2*θ*) and a scan step time of 100 s.

SEM was performed using a JSM-7100F electron microscope from JEOL, (Tokyo, Japan). The samples were sputter-coated with carbon (20 nm) using a QISOT ES Sputter Coater (Quorum Technologies, Lewes, UK) and fixed with carbon tape on an SEM sample holder. Micro-computed tomography (μ-CT) imaging was performed with a phoenix v|tome|x L 240 device, developed by GE Sensing & Inspection Technologies. The samples were analyzed using a 180 kV/15 W nanofocus tube. Scanning parameters were set as follows: voltage 60 kV, current 210 μA, projections 1800, average 3, skip 1, timing 500 ms, detector sensitivity 1, and voxel size 3 μm. After the scanning process, 3D data sets were evaluated with VG Studio Max 2.2, and the Wall thickness analysis module was used to determine the composite film thickness.

Contact angles were measured on a DSA30 (Krüss), with the software package Drop Shape Analysis 4. A drop of 5 μL was placed on the sample at 100 μL min^−1^. Thereafter, the drop volume was increased to 20 μL at 15 μL min^−1^ and subsequently decreased again to 5 μL again at 15 μL min^−1^. At least ten different measurements were obtained for every sample. Data points were collected every 5, 30, and 60 s with the “tangent 2” method on the right and left sides of the drop. ANOVA test was used for the comparison of the data using Origin Lab software.

IR spectra were recorded on a Nicolet^TM^ is50 Fourier transform infrared spectrometer (Thermo Scientific, Waltham, MA, USA) equipped with an IR source, KBr beamsplitter, and DTGS detector using a single-reflection ATR accessory with the diamond crystal. Spectra were recorded with a resolution of 4 cm^−1^ using the Thermo Scientific OMNIC^TM^ software. XPS was performed with a Nexsa (Thermofisher) using AlK_α_ radiation operating at 72 W, pass energy of 200 eV, and a spot size of 400 μm with an integrated flood gun. The Standard Lens Mode in CAE Analyser Mode was used with an energy step size of 1 eV for the survey spectrum (30 passes). Samples were analyzed after the surface was treated with Ar-clusters (1000 Ar atoms, 6000 eV, 60 s). High-resolution spectra (step size 0.1 eV) of the single elements were acquired with 30 passes at a pass energy of 50 eV. Peak analysis was performed with Thermo Avantage v5.9925 Build 06702. Series of IR and XPS spectra were analyzed using the MCR method to identify the main spectral components. The details of this method are described below.

Absorption spectra of hybrid colloidal dispersion, as well as the polymer nanocomposites, were measured using a CARRY 5000 spectrophotometer (Agilent, Santa Clara, CA, USA) in the range of 200–800 nm. Liquid samples were measured using 10 mm quartz cuvettes (Hellma Analytics, Mullheim, Germany). Spectra of films and PCL nanocomposites were recorded using UV-Vis-NIR Praying Mantis Diffuse Reflectance Accessory (Agilent, Santa Clara, CA, USA) and the spectra were corrected by subtracting that of the blank sample. All the spectra were recorded at room temperature.

Emission and excitation spectra of selected samples containing MB were recorded using a FluoroLog 3 spectrophotometer (Horiba Jobin Yvon, Kyoto, Japan) in a front-face setup. The films and PCL nanocomposites were analyzed using the J1933 Solid Sample Holder of Horiba. The emission spectra were recorded using the excitation at 590 nm. Fluorescence quantum yields were measured using Quanta-Phi cell (Horiba Jobin Yvon) using the excitation at 580 nm in the range of emission of 590–775 nm. The measured signals were corrected for the wavelength dependence of the spectrofluorometer, effects of optics, and the integrating sphere. The excitation beam was measured using the blank sample holder composed of Spectralon^®^. A Neutral-density filter was used to reduce the photon flux of an excitation beam and the transmittance of the filter was measured to get a correct area balance factor. Ten emission spectra were recorded for each sample and the mean spectrum was calculated. The background spectrum measured for an empty container was subtracted to get a pure emission signal which was additionally smoothed by the Savitzki–Golay method to remove spectral noise.

Chemometric analysis based on the method of multivariate curve resolution—alternating least square (MCR) was applied to distinguish individual signals from those of background and other components [44]. This mathematical iterative method is capable to decompose a complex set of data in a single analysis to get individual signals from chemical components and a signal background without the need for any information on the compounds present in the analyzed samples. The method is based on the multi-component Lambert–Beer law according to the relationship: **X = CS^T^**.

**X** is the spectrum matrix of the sample set, **C** is the matrix of vectors representing the arbitrary concentration profiles of the respective spectral components, and **S^T^** is the corresponding spectra matrix. The MCR method was used to obtain information from XPS and IR spectra. To get correct results the spectra of simple precursors were included in the spectral matrices. The algorithm of Unscrambler software (Camo, Norway) was used applying constraints of non-negativity of both spectra and concentrations.

## 4. Conclusions

Conventional polymer composites have particles more or less uniformly dispersed in the polymer matrix. In this work, materials with particles located only on or close to the polymer surface were prepared. The materials were prepared by the fusion of molten polycaprolactone polymer upon contact with a thin layer of a modified layered silicate. The surface parameters of such materials are directly related to the properties of the particles and less to polymer properties. On the other hand, the polymer bulk remaining without the filler particles does not lose most of its original properties. By controlling the thickness of the silicate precursor film, it is possible to adjust the properties and amount of the silicate particles on the modified polymer surface. The fusion of the two phases was successfully realized. Materials with methylene blue bound directly to silicate particles were also tested and some of the composites exhibited fluorescence. This can be important for future development of photoactive, photosensitizing, and possibly also photodisinfection materials. To achieve surfaces of superior activity, it is necessary to further optimize these materials, which will soon be the subject of further research.

## Figures and Tables

**Figure 1 ijms-23-09166-f001:**
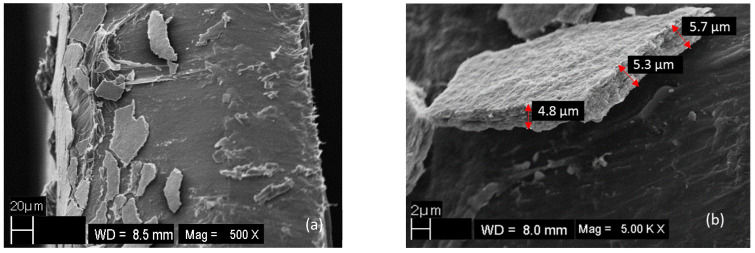
SEM and X-ray tomography images and the film thickness distribution of PCL/MB10 composite. The sections perpendicular to the surface by SEM (**a**,**b**) and an image of cross-section obtained by X-ray tomography (**c**), and the statistical distribution of the thickness values of the composite film (**d**). The most frequent thickness was estimated by the nonlinear fitting of the data using an asymmetric double Sigmoidal function (dotted line) using Origin Lab software.

**Figure 2 ijms-23-09166-f002:**
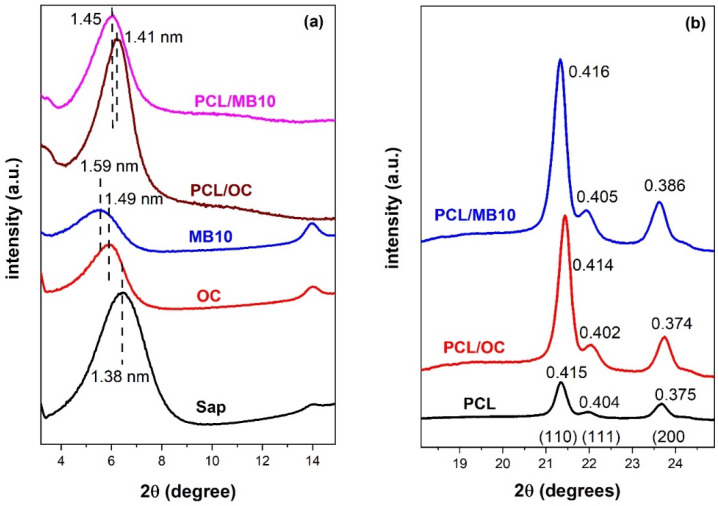
X-ray diffraction patterns of (**a**) the films of Saponite (Sap), organoclay (OC), organoclay with MB (MB10) deposited on PTFE filter membranes, and related composites PCL/OC and PCL/MB10. (**b**) Details on the pattern at longer angles showing the reflections due to the polymer phase (110), (111), and (200). The numbers labeling the reflection peaks show *d* values (nm).

**Figure 3 ijms-23-09166-f003:**
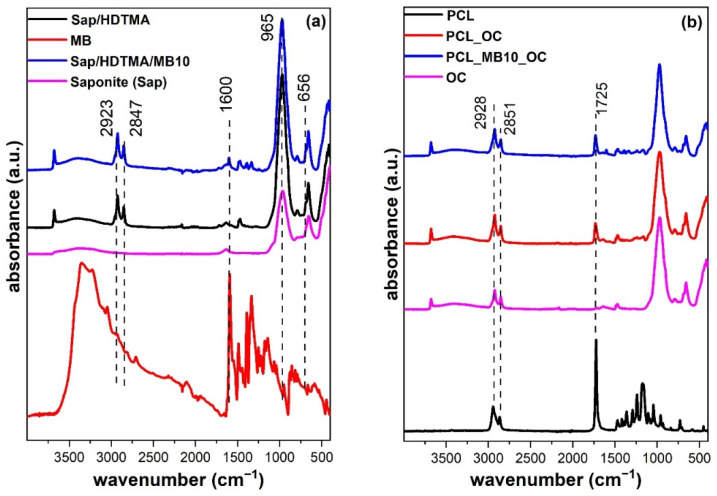
ATR spectra of saponite (Sap), methylene blue (MB), organoclay (OC, MB10) (**a**), polycaprolactone (PCL) and its composites (PCL/OC, PCL/MB10) (**b**).

**Figure 4 ijms-23-09166-f004:**
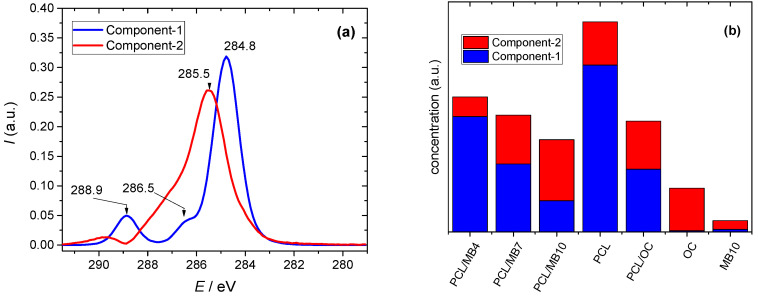
Spectra (**a**) and fractions (**b**) of the components calculated by multivariate curve resolution—alternating least squares of the X-ray photoelectron spectra in the C1s region.

**Figure 5 ijms-23-09166-f005:**
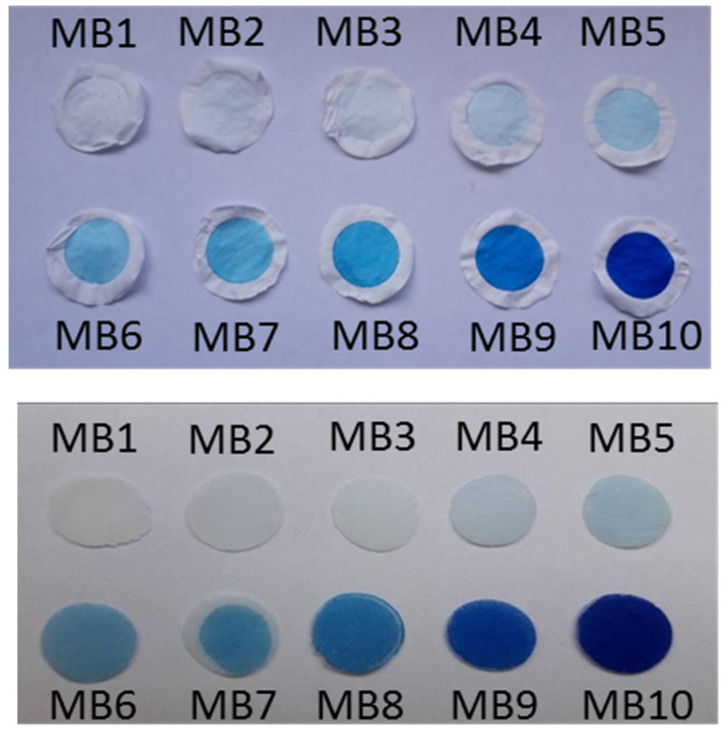
Photographs of the films of organoclays functionalized with methylene blue deposited on filtration membranes (**top**) and membranes of corresponding composites with the polymer (**bottom**).

**Figure 6 ijms-23-09166-f006:**
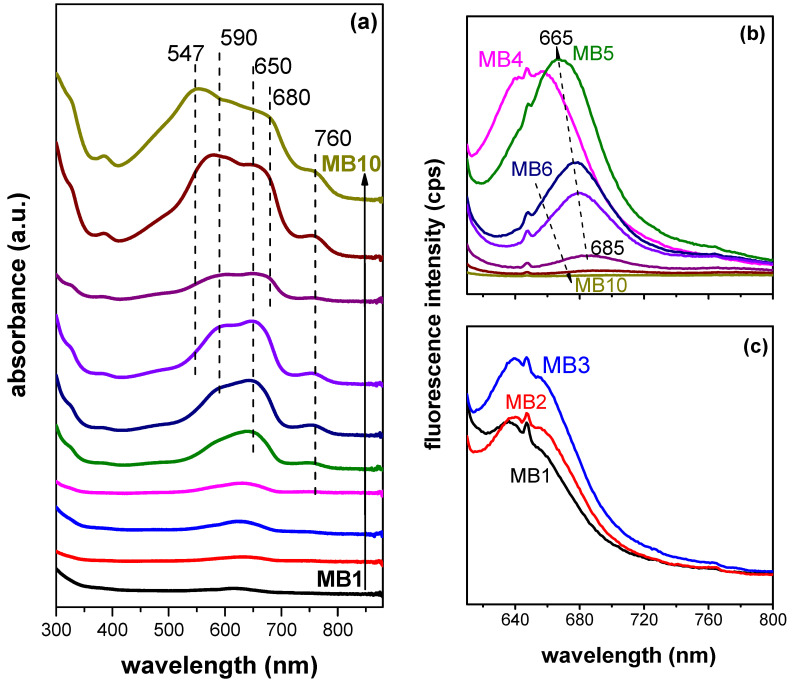
UV-Vis absorption (**a**) and fluorescence spectra (**b**,**c**) of composite films functionalized with methylene blue. The scales of Figure 6b,c are the same.

**Figure 7 ijms-23-09166-f007:**
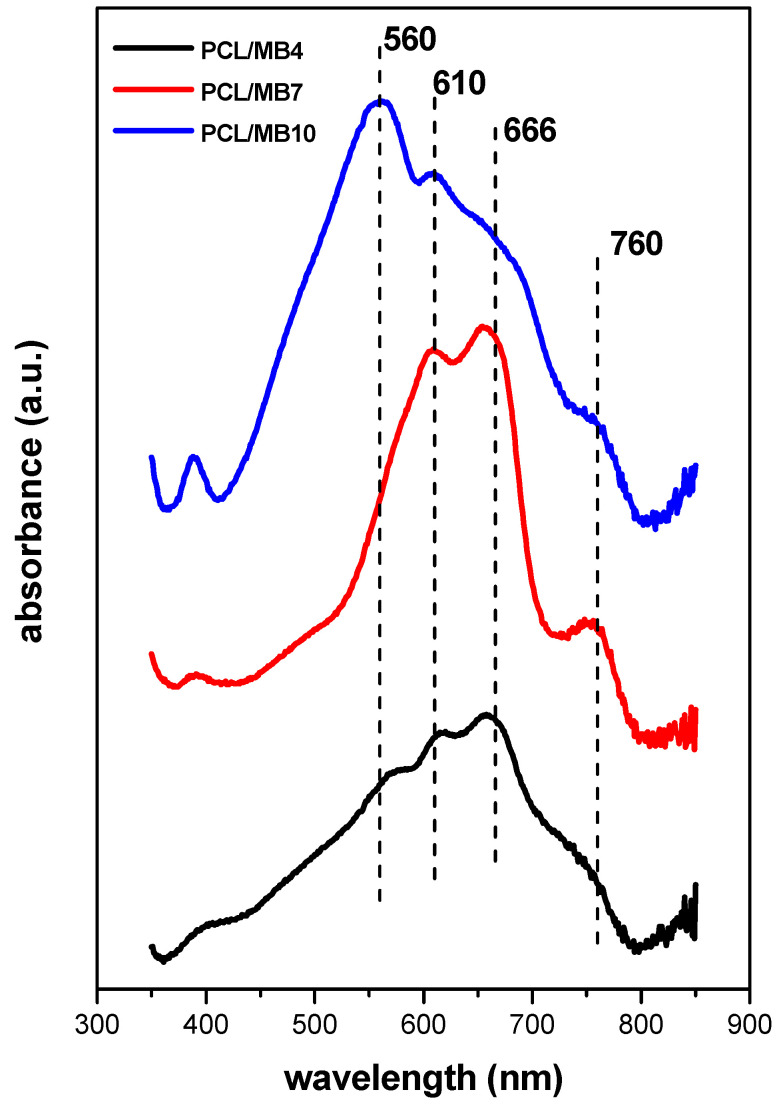
UV-Vis spectra of PCL discs modified with composite films functionalized with methylene blue.

**Figure 8 ijms-23-09166-f008:**
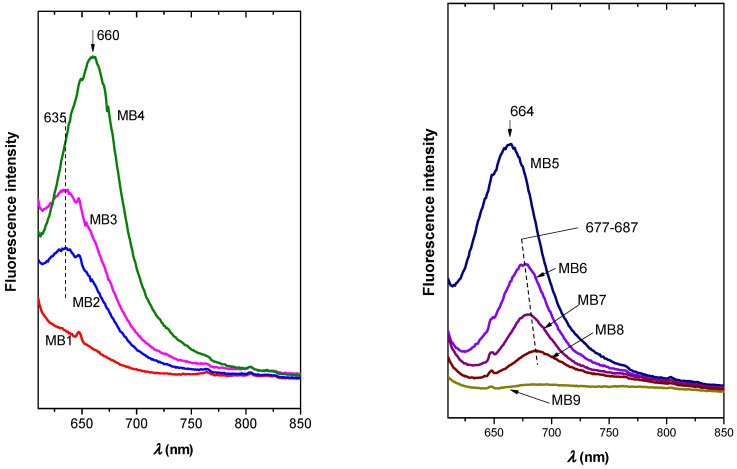
Fluorescence spectra of the composites of PCL discs with MB1-MB9 composite films.

**Table 1 ijms-23-09166-t001:** Mean values and standard deviations of water contact angles of selected samples. The values are from over 30 repeated measurements.

Organoclay	Water Contact Angle/°	Composite	Water Contact Angle/°
OC	66 ± 10	PCL	62 ± 5
MB4	63 ± 9	PCL/MB4	65 ± 6
**MB10**	**47 ± 5**	PCL/MB10	65 ± 9

The sample of significantly lower hydrophobicity is highlighted.

**Table 2 ijms-23-09166-t002:** The occurrence of spectral components calculated by multivariate curve resolution—alternating least squares in the infrared spectra of simple and multicomponent samples.

	PCL_MCR_	OC_MCR_	MB_MCR_
OC	0	**98**	0
MB10	0	94	5.2
MB	2.7	1.1	**100**
PCL	**100**	0	0.8
PCL/OC	19	**100**	1.4
PCL/MB10	18	83	2.2

The values were normalized to the highest signal of a component occurring in the series and expressed in %. Large bold numbers denote the maximal values of ~100%.

**Table 3 ijms-23-09166-t003:** Final compositions of the colloidal dispersions of organoclays MB1-MB10 used for the preparation of organoclay films.

Sample	MB1	MB2	MB3	MB4	MB5	MB6	MB7	MB8	MB9	MB10
*c*_MB_ /10^−5^ mol L^−1^	0.02	0.04	0.05	0.1	0.2	0.4	0.5	1	2	4
*n*_MB_/*m*_Sap_/ 10^−6^ mol g^−1^	0.5	1	1.25	2.5	5	10	12.5	25	50	100

The final concentrations of HDTMA (*c*_HDTMA_) and Sap (*c*_Sap_) were always 3.2 × 10^−4^ mol L^−1^ and 0.4 g L^−1^, respectively.

## Data Availability

Data are available from authors upon request.

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
