# Peer review of "Modified Polymer Surfaces: Thin Films of Silicate Composites via Polycaprolactone Melt Fusion"

_ijms, 2022, doi:10.3390/ijms23169166_

Round 1
Reviewer 1 Report
In this study, a composite thin film is coated on the surface of a polymer by the means of a new method of formation. The thin layer comprises of saponite particles modified via alkylammonium cations as well as being functionalized by the means of methylene blue. A framework was considered to interconnect matrix and saponite. In addition, the surface morphology is explored via experimental observation. According to above facts, the merits of manuscript cannot be ignored. The topic is of great magnificence and is well discussed about within the text. So, in general, manuscript is already to be considered for possible publication in International Journal of Molecular Sciences. However, to strengthen the paper, the writers are invited to reflect their response to bellow comments to the reviewer so that the decision-making process is being simpler.
1. Although the reviewer is aware of the central core of the research, i.e. related to coaptation of thin composite layer on polymer, it is much better to explain more practical aspects of synthesization of polymer nanocomposites (PNCs) in the Introduction section for the sake of completeness.
2. The authors did not pay attention to stiffness reinforcing mechanism in PNCs which can be dramatically affected by several destructive phenomena such as entanglement of nanoparticles in clusters, non-ideality of the nanoparticles, non-ideality of the interfacial bonding between matrix and nanoparticles, possibility of existence of pores and voids in the microstructure and so forth. Since the abovementioned issues are of incredible importance in the design process, the authors are invited to bring detailed explanations in this regard. Useful data in this are can be found by referring to the most recent book published by Elsevier in this field (“Mechanics of Multiscale Hybrid Nanocomposites”, 1st ed., 2022, eBook ISBN: 9780128231623).
3. More data about functionalization is required. As the authors know, although functionalization enhances cross-linking, there exist difference between efficiency of various kinds of functionalization. This reality needs to be mentioned in the text so that junior readers are not misunderstood.
4. As I saw in the text, Materials and Methods are presented after Results and Discussion. The reviewer strongly recommends to change the order so that the continuity is kept within the manuscript. The best way to write experiment-assisted articles is to write them in the same order as the process of data generation was fulfilled by the authors.
5. The Conclusion section can be decreased in size. In this section, it is only enough to re-discuss the most important highlights by giving more practical insights to the readers.
The results are well interpreted and this point was deliberately mentioned by the reviewer to thank the authors. In most experiences of mine in reviewing manuscripts, the authors only report the data and do not pay attention to the physics of problem; but here, the explanations were sufficient. So, I want to thank you for your structured discussion in this section.
Author Response
- Although the reviewer is aware of the central core of the research, i.e. related to coaptation of thin composite layer on polymer, it is much better to explain more practical aspects of synthesization of polymer nanocomposites (PNCs) in the Introduction section for the sake of completeness.
Response to the reviewer’s comments:
The first paragraph of the Introduction section was partly expanded to include more sentences on the synthesis of polymer nanocomposites, in particular on the melt intercalation which is the method used also in this study.
- The authors did not pay attention to stiffness reinforcing mechanism in PNCs which can be dramatically affected by several destructive phenomena such as entanglement of nanoparticles in clusters, non-ideality of the nanoparticles, non-ideality of the interfacial bonding between matrix and nanoparticles, possibility of existence of pores and voids in the microstructure and so forth. Since the abovementioned issues are of incredible importance in the design process, the authors are invited to bring detailed explanations in this regard. Useful data in this are can be found by referring to the most recent book published by Elsevier in this field (“Mechanics of Multiscale Hybrid Nanocomposites”, 1sted., 2022, eBook ISBN: 9780128231623).
Response to the reviewer’s comments:
The reviewer is correct that the structure of the composite significantly affects the mechanical properties and practical parameters of a polymer important for technical applications. The formation of the composite and the presence of nanoparticles can sometimes lead to an improvement in some parameters, while some properties are often adversely affected. The systems prepared and studied in this work represent polymer membranes with surfaces modified with functionalized layered silicate particles. Since the modification represents a film with a thickness of only a few micrometers on the surface of the membrane, we do not expect that the properties of the bulk polymer (beyond the film on the surface) would change significantly. On the other hand, the mechanical properties of the thin layer probably differ significantly from the bulk of the pure polymer. In this work, we have not yet dealt with mechanical properties, as this is a specific research area requiring a detailed study of the size equal to a few specialized papers. There is no room to expand the size of this manuscript in this direction. On the other hand, we plan to investigate this type of material in this direction. It is due to the advances in our research to get desired structure of the composite layer on the polymer surface.
- More data about functionalization is required. As the authors know, although functionalization enhances cross-linking, there exist difference between efficiency of various kinds of functionalization. This reality needs to be mentioned in the text so that junior readers are not misunderstood.
Response to the reviewer’s comments:
Functionalization of the polymer itself was not implemented in this work. On the other hand, the silicate particles were subjected to modification with alkylammonium cation and functionalization with methylene blue. Methylene blue is formed by cations that bind very strongly to the surface of negatively charged silicate particles. We do not expect any significant interaction of the dye cations with the polymer.
The procedures of functionalization of silicate using methylene blue are described in detail in the Method section. Since there is strong adsorption of dye cations on the silicate surface, the amount of adsorbed dye was determined by the amount of added dye. The properties of the materials, whether they were precursors in the form of films on PTFE membranes or the final PCL membranes, are described in detail in the section focused on the optical properties of the materials.
- As I saw in the text, Materials and Methods are presented after Results and Discussion. The reviewer strongly recommends to change the order so that the continuity is kept within the manuscript. The best way to write experiment-assisted articles is to write them in the same order as the process of data generation was fulfilled by the authors.
Response to the reviewer’s comments:
We originally intended to follow the order as recommended by the reviewer. However, the guidelines of the journal strictly state that the Materials and Methods section should be at the end of the text, after the Results and discussion section. I have verified this issue with the editors and the rule is mandatory for papers published in the International Journal of Molecular Sciences. However, some additional text was added to some parts of the Results and discussion sections, so that the continuity of the experiments would be understood by a reader.
- The Conclusion section can be decreased in size. In this section, it is only enough to re-discuss the most important highlights by giving more practical insights to the readers.
Response to the reviewer’s comments:
This section was partially shortened, so as not to duplicate the parts discussed in the previous section. The conclusions regarding the contribution of the work to the future development of materials of this type, as well as a brief presentation of the direction of future research, remained.
The results are well interpreted and this point was deliberately mentioned by the reviewer to thank the authors. In most experiences of mine in reviewing manuscripts, the authors only report the data and do not pay attention to the physics of the problem; but here, the explanations were sufficient. So, I want to thank you for your structured discussion in this section.
The authors would like to thank the reviewer for a fair and constructive review.
Reviewer 2 Report
This manuscript intends to prepare polycaprolactone membranes with a surface modified with an organoclay layer. The authors claim that the study provides directions for future research aimed at the development of composite materials with photosensitizing, photodisinfection, and antimicrobial surfaces.
1) What is the research hypothesis. It should be written in the introduction section.
2) The results of this study must be compared with those of the published papers.
3) Authors claim that "These results proved that in this way it is possible to prepare polymer nanocomposites whose surfaces have photoactive, photosensitizing, and possibly also photodisinfection properties". The authors must investigate the application of the prepared polymeric nanocomposites.
Author Response
1) What is the research hypothesis. It should be written in the introduction section.
Response to the reviewer's comments:
The hypothesis was added in the Introduction section. It is based on the feasibility to prepare hybrid systems with a locally increased concentration of dispersed particles located on or near the surface of the polymer. The way how to achieve was designed using the fusion of polymer melt with the thin film of layered silicate.
2) The results of this study must be compared with those of the published papers.
Response to the reviewer's comments:
Until now, no paper has been published using the same type of modification of the polymer surface described in this work. A method is an original approach and there is not much to compare. What is known, however, are conventional composites of polycaprolactone with organoclays, properties of precursor materials such as PCL, organoclays, and their complexes with methylene blue, which have been described in several publications. Therefore, results on the structure or properties of composite materials in each of the subsections are analyzed by comparing with the results of older published studies. Relevant literature was studied in detail before analyzing measured data. For example, the results in reference [5] were used to evaluate surface hydrophobicity, references [18-27] for crystalline phase structure, [28-32] for chemical composition and the presence of functional groups, and [33-39] for optical properties related to the presence of methylene blue.
3) Authors claim that "These results proved that in this way it is possible to prepare polymer nanocomposites whose surfaces have photoactive, photosensitizing, and possibly also photodisinfection properties". The authors must investigate the application of the prepared polymeric nanocomposites.
Response to the reviewer's comments:
The antimicrobial properties themselves were not the goal of this work. The objective was the implementation of the method to achieve polymer surface modification as described in the Introduction section. Of course, one of the future goals is the use of such materials as antimicrobial surfaces.
Nevertheless, basic pilot microbiological tests on the specimens prepared in this work were carried out, but the results showed only a small effect of surface modification on the reduction of biofilm growth (~ 50%). With other systems, we could achieve a 100- to a 1000-fold reduction of the biofilm growth. The problem is the photoactivity reduction as the surface MB concentration increased, probably due to molecular aggregation. Photoactivity is necessary for the effective functioning of the photosensitizer. We have decided not to include the pilot results in this work yet and to devote a specialized study to improving the antimicrobial properties which are carried on now. The results of the current investigation are promising but require some significant changes and novel approaches. We believe it will be part of another manuscript shortly.